# Model-Based Characterization of *E. coli* Strains with Impaired Glucose Uptake

**DOI:** 10.3390/bioengineering10070808

**Published:** 2023-07-05

**Authors:** Niels Krausch, Lucas Kaspersetz, Rogelio Diego Gaytán-Castro, Marie-Therese Schermeyer, Alvaro R. Lara, Guillermo Gosset, Mariano Nicolas Cruz Bournazou, Peter Neubauer

**Affiliations:** 1Chair of Bioprocess Engineering, Institute of Biotechnology, Technische Universität Berlin, Ackerstr. 76, 13355 Berlin, Germany; n.krausch@tu-berlin.de (N.K.); l.kaspersetz@tu-berlin.de (L.K.); schermeyer@tu-berlin.de (M.-T.S.); mariano.n.cruzbournazou@tu-berlin.de (M.N.C.B.); 2Departamento de Ingeniería Celular y Biocatálisis, Instituto de Biotecnología, Universidad Nacional Autónoma de México, Cuernavaca 62209, Mexico; gaytandiego@gmail.com (R.D.G.-C.); guillermo.gosset@ibt.unam.mx (G.G.); 3Departamento de Procesos y Tecnología, Universidad Autónoma Metropolitana, Mexico City 05348, Mexico; alara@cua.uam.mx; 4DataHow AG, 8050 Zurich, Switzerland

**Keywords:** laboratory automation, model-based, strain characterization, high throughput

## Abstract

The bacterium *Escherichia coli* is a widely used organism in biotechnology. For high space-time yields, glucose-limited fed-batch technology is the industry standard; this is because an overflow metabolism of acetate occurs at high glucose concentrations. As an interesting alternative, various strains with limited glucose uptake have been developed. However, these have not yet been characterized under process conditions. To demonstrate the efficiency of our previously developed high-throughput robotic platform, in the present work, we characterized three different exemplary *E. coli* knockout (KO) strains with limited glucose uptake capacities at three different scales (microtiter plates, 10 mL bioreactor system and 100 mL bioreactor system) under excess glucose conditions with different initial glucose concentrations. The extensive measurements of growth behavior, substrate consumption, respiration, and overflow metabolism were then used to determine the appropriate growth parameters using a mechanistic mathematical model, which allowed for a comprehensive comparative analysis of the strains. The analysis was performed coherently with these different reactor configurations and the results could be successfully transferred from one platform to another. Single and double KO mutants showed reduced specific rates for substrate uptake q_Smax_ and acetate production q_Apmax_; meanwhile, higher glucose concentrations had adverse effects on the biomass yield coefficient Y_XSem_. Additional parameters compared to previous studies for the oxygen uptake rate and carbon dioxide production rate indicated differences in the specific oxygen uptake rate q_Omax_. This study is an example of how automated robotic equipment, together with mathematical model-based approaches, can be successfully used to characterize strains and obtain comprehensive information more quickly, with a trade-off between throughput and analytical capacity.

## 1. Introduction

*Escherichia coli* is one of the most prominent organisms used for the biotechnological production of recombinant proteins, nucleic acids, and various metabolites [1]. The reasons for this are the fast growth rate, the well-studied genome, the simple process conditions, and a variety of protocols for the genetic engineering of the organism [2]. Moreover, advances in metabolic engineering in combination with high-throughput technologies have facilitated the creation of large strain libraries. However, the sheer variety of possible combinations of strains and process conditions leads to a bottleneck in strain characterization and necessitates consistent development at all stages [3].

Under aerobic and non-limiting glucose conditions, *E. coli* grows at high growth rates (μ) and has correspondingly high rates of substrate consumption (q_S_). This leads to a metabolic imbalance between the fluxes through glycolysis and the tricarboxylic acid cycle (TCA). This phenomenon is called overflow metabolism; it results in the secretion of several organic acids, especially acetate [4,5,6]. The production of acetate can inhibit further growth and represents a loss of carbon; thus, acetate formation should be prevented [7]. Previous studies have shown that knockout strains with one or more of the genes related to the uptake and processing of glucose (Figure 1) have significantly lower specific growth rates than the wild-type (WT); however, they also have a significantly lower acetate production rate and are interesting host strains as microbial cell factories [8,9,10]. In this regard, batch cultivation can be more advantageous than fed-batch cultivation, allowing for high productivity and reduced overflow metabolism by using such mutant strains, which limit glucose uptake [11,12].

Fuentes et al. [9] carried out a characterization of *E. coli* W3110 derivatives with sugar uptake systems impairments. WG, WGM, and WGP were among these strains with single or double mutations. WG is an isogenic derivative of strain W3110, in which the genes for the EII glucose complex were eliminated. WGM and WGP are derivatives of WG, with the EII mannose complex additionally impaired in the former. In contrast, strain WGP has a deletion of the *galP* gene. WG and WGP showed a similar growth rate (77% of WT) and reduced specific acetate production (34% of WT). WGM had a lower specific growth rate (55% of WT); however, it showed no acetate formation and, thus, a higher biomass yield on glucose. These strains appear to be interesting hosts for biotechnological applications. Their characterization has mostly been performed following traditional screening in microtiter plates (MTPs) or shake flasks [11], where promising candidates are transferred to lab-scale reactors allowing for better process control [13]. However, the information obtained from MTPs or shake flasks is limited because these systems are constrained in their monitoring and especially in their process control [14]. Therefore, miniature bioreactor systems operated in Liquid Handling Stations (LHSs) offer a trade-off between throughput and process control that is comparable to lab-scale reactors [15,16,17]. In combination with the use of mathematical models, the physiological properties of bacterial strains can be more efficiently determined. In an effort to address this challenge and to reduce the risk of failure during scale-up, high-throughput (HT) mini bioreactor (MBR) systems based on model-based operation strategies have been developed [18] and extended to conditional screening experiments [19]. Such a combination allows for determining cell-specific parameters, such as yields and specific rates under process-relevant conditions. Nevertheless, the limitations of these MBRs remain, concerning individual process control, sampling, measurement frequency, online process analytical tools, and relatively low oxygen transfer rates (OTRs) [20]. In contrast, laboratory-scale reactors offer higher sampling volumes but lack a sufficient degree of automation. Morschett et al., addressed this issue by integrating a parallel laboratory scale reactor system in a LHS for automated cultivation workflows [17]. Additionally, these systems can be easily equipped with a variety of online process analytical tools. Off-gas analysis, for instance, is a non-invasive method, which is available up to an industrial scale, providing insights into respiration-related parameters [21]. Oxygen uptake rates (OURs) and carbon dioxide production rates (CPRs) can be determined from mass-balance equations. Moreover, such systems offer the possibility to implement simple and robust state estimators for biomass from an early stage [22,23]. For a comprehensive characterization of strains under process-relevant conditions, these robotic cultivation systems must not be seen as single entities. Digital laboratory infrastructure allows for the sharing of a common modeling framework; meanwhile, mobile robots can physically connect robotic cultivation systems of different scales or guarantee access to the same analytical devices [24].

Herein, we present the model-based characterization of three isogenic *E. coli* strains with gene knockouts (KOs) in the phosphotransferase system (PTS) and their impaired glucose uptake capabilities in a high-throughput robotic facility. The experiments were conducted in a high-throughput robotic facility, which included a small-scale system, a mobile robotic lab assistant [24], and a parallel mini-bioreactor system [16] with a common data infrastructure. It has already been successfully demonstrated, that this platform provides a versatile and efficient means of rapidly screening strains and processes [25]. After performing the initial screening experiments in MTPs, the strains were cultivated in our high-throughput facility in 10 mL MBRs and 100 mL small-scale bioreactors under process-relevant conditions. For each strain, a parameter estimation was performed to characterize and compare the strains and highlight the advantage of a common data infrastructure.

The approach is outlined in Figure 2. With this approach, we demonstrate the added value of an automated and digitally connected cultivation platform with different cultivation stages that enable comprehensive strain screening. The platform approach facilitates the model-based characterization of three *E. coli* KO strains with interesting properties; they are able to grow efficiently under batch conditions with high glucose levels.

## 2. Materials and Methods

### 2.1. Strain and Cultivation Conditions

All experiments were carried out with *E. coli* KO strains, which were obtained from a previous study [9]. The genotypes are described in Table 1.

All chemicals were obtained from Roth (Carl Roth GmbH, Karlsruhe, Germany), Merck (Merck KgaA, Darmstadt, Germany), or VWR (VWR International, Radnor, PA, USA), if not stated otherwise. TY-medium contained: 16 g L^−1^ bacto tryptone (Becton Dickinson, Franklin Lakes, NJ, USA), 10 g L^−1^ bacto yeast extract (Biospringer, Maisons-Alfort, France) and 5 g L^−1^ NaCl. The main medium consisted of a mineral salt medium (MSM), containing, in g L^−1^: 2Na_2_SO_4_, 2.468(NH_4_)_2_SO_4_, 0.5NH_4_Cl, 14.6K_2_HPO_4_, 3.6NaH_2_PO_4_·2H_2_O, 1(NH_4_)_2_-H-citrate, and 1 mL antifoam (Antifoam 204, Sigma-Aldrich, St. Louis, MO, USA). The medium was supplemented with 2 mL L^−1^ trace elements solution, (1 M) MgSO_4_ solution, and 0.005 g L^−1^ thiamine hydrochloride and was adjusted to pH 7. The trace element solution comprised, in g L^−1^: 0.5CaCl_2_·2H_2_O, 0.18ZnSO_4_·7H_2_O, 0.1MnSO_4_·H_2_O, 20.1Na-EDTA, 16.7FeCl_3_·6H_2_O, 0.16CuSO_4_·5H_2_O, 0.18CoCl_2_·6H_2_O, 0.087Na_2_SeO_3_ (Alfa Aesar, Haverhill, MA, USA), 0.12Na_2_MoO_4_·2H_2_O, and 0.725Ni(NO_3_)_2_·6H_2_O.

TY-medium (15 mL) was directly inoculated with 100 μL of cryo culture and cultured in a 125 mL Ultra-Yield™ flask sealed with an AirOtop™ enhanced flask seal (both from Thomson Instrument Company, Oceanside, CA, USA) for 7 h at 37 °C and 220 rpm in an orbital shaker (25 mm amplitude, Adolf Kühner AG, Birsfelden, Switzerland). The second preculture was set to an optical density at 600 nm (OD_600_) of 0.25 and was cultured in 50 mL of MSM with 25% (v v^−1^) TY-medium in an Ultra-Yield™ flask sealed with an AirOtop™ enhanced flask seal under the same conditions. After 12 h, appropriate volumes of the preculture were used to inoculate the main culture to an OD_600_ of 0.5.

### 2.2. HT Bioprocess Development Facility

The HT bioprocess facility consisted of two automated cultivation platforms with working volumes ranging from 10 mL (described in Section 2.2.2) to 150 mL (described in Section 2.2.3), a LHS (described in Section 2.2.2), and a HT-analyzer (described in Section 2.2.5) for the atline analysis of glucose, acetate, and OD_600_. The spatially separated platforms were physically connected by a mobile robotic lab assistant (Section 2.2.4), allowing for automated sample transport.

#### 2.2.1. MTP Cultivation

For the initial growth characterization, strains were cultivated with either 5, 10, or 20 g L^−1^ of glucose in 24-well OxoDish^®^ (PreSens, Regensburg, Germany) MTPs in 1 mL medium, as described above, with an OD_600_ of 0.25. Dissolved oxygen tension (DOT) was continuously measured with optodes at the bottom of the plates. The MTPs were cultivated at 37 °C in an orbital shaker at 250 rpm and a shaking amplitude of 50 mm.

#### 2.2.2. 2mag Cultivation Platform

The first validation experiments were conducted on a high-throughput bioprocess development platform. The platform consisted of two liquid handling stations. A bioreactor 48 MBR system (2mag AG, München, Germany) was embedded in one of the LHSs. The working volume of the MBRs was 10 mL. Each of the reactors was equipped with fluorometric sensors to determine pH and DOT. Feeding and sampling were performed automatically via the LHS in a predefined time frame. Samples were then automatically transferred to the second LHS, where OD_600_ was measured and glucose and acetate concentrations were determined via enzymatic assays. The reader can be referred to [16] for a detailed description of the platform, the sampling, and the feeding procedures.

The cultures were run in the parallel stirred tank mini bioreactors at 37 °C and pH was controlled at 7.0 with 3.5 M NH_4_OH. The main cultures were started as 10 mL batch cultures with their OD_600_ set to 0.5 and initial glucose concentrations of 5 or 10 g L^−1^. After the batch phase ended, a pulse of glucose was automatically given, either once in the case of a 10 g L^−1^ initial glucose concentration, or twice in the case of a 5 g L^−1^ initial glucose concentration. The system was aerated with compressed air at 5 L min^−1^ and the stirring speed was kept constant at 2400 rpm for every reactor.

#### 2.2.3. BioXplorer Cultivation Platform

The BioXplorer 100 (H.E.L group, London, UK), suitable for eight parallel cultivations in glass stirred tank reactors (STRs), was applied for the second validation experiments. The cultivation system had two main components: (1) a heating and cooling block (PolyBlock) that holds the vessels and (2) the control unit. The system was equipped with pH and DOT sensors (both from AppliSens, Applikon Biotechnology B.V., Delft, The Netherlands), three peristaltic pumps, and a mass flow controller for each vessel. The PolyBlock was integrated into a LHS (Tecan EVO 150, Tecan Group, Männedorf, Switzerland), while the control unit was placed on the left-hand side of the LHS. Additionally, off-gas analyzers (BlueVary, BlueSens, Herten, Germany) measuring the concentrations of carbon dioxide, oxygen, absolute humidity, and pressure were integrated into each vessel. The reader can be referred to [24] for a detailed description of the platform and sampling procedures.

Cultures were run in the parallel glass stirred tank reactors (STRs), equipped with one Rushton-type impeller, at 37 °C. The pH was controlled at 7.0 with 8% (v v^−1^) NH_4_OH via the WinISO control software (H.E.L group, London, UK). The main cultures were started as 90 mL batch cultures with their OD_600_ set to 0.5 and initial glucose concentrations of 20 g L^−1^. The cultivations were started with aeration with a flow rate of 0.22 vvm and a stirring speed of 1000 rpm. The air-flow rate and stirring speed were increased to 0.66 vvm after 2.75 h and to 1.66 vvm and 1500 rpm after 6.8 h.

#### 2.2.4. Mobile Robotic Lab Assistant

The mobile robotic lab assistant (Astechproject Ltd., Runcorn, UK), consisting of a driving platform MIR100, a robotic arm URE5 equipped with a 2-finger gripper, and a 3D camera, was used for automated sample transport. The driving platform was equipped with two 3D cameras and two laser scanners for navigation and safety. The 2-finger gripper could pick up MTPs in either the portrait or landscape position. Two MTPs could be stored on the deck and additional four plates could be stored at the back, on a shelf of the platform. The laboratory map was taught, including device positions for the charging station and additional way-points. The BioXplorer, 2mag, and the high-throughput analyzer (described in Section 2.2.3, Section 2.2.2, and Section 2.2.5, respectively) were labeled with a marker (ApriTag) and the corresponding positions for the MTPs were taught from a right-handed position. The corresponding functions for picking and placing a plate were set to linear movements (moveL) in PolyScope 3.2 (Universal Robots, Odense, Denmark). The movement velocity of the robot (0.8 m s^−1^), as well as the operating height, were restricted due to safety reasons. All necessary functions for setting up and executing a workflow were controlled via a SiLA2 interface.

#### 2.2.5. High-Throughput Metabolite Analysis

The atline analysis for glucose and acetate was conducted by a HT-analyzer (Cedex BioHT, Roche Diagnostics GmbH, Mannheim, Germany) equipped with a rack suitable for 96-MTPs and an opening in the front lid. The following test kits were calibrated and validated with the corresponding controls according to the manual and prior to use: Glucose Bio HT and Acetate V2 Bio HT (both from Roche Diagnostics GmbH, Mannheim, Germany).

### 2.3. Data Handling

All online, atline, and offline measurements, set points, control inputs, and sampling steps were stored in a central SQL database. This allowed for process control and parameter estimation in an online and adaptive manner, as well as data transfer between cultivation platforms.

### 2.4. Mathematical Modeling Tools

The modeling framework was based on a mechanistic *E. coli* model with glucose partitioning, overflow metabolism, and acetate re-cycling and has been proven for strain characterization [19,26]. The system was formulated as a differential-algebraic system of equations able to describe the dynamic changes in important state variables, such as glucose, acetate, biomass, and DOT. The parameters of the model were obtained by fitting the model to the experimental data. A more detailed description of the underlying model and the functioning of the framework can be found in [27].

### 2.5. Off-Gas Analysis

The OUR and CPR can be described via Equations (1) and (2) as follows:(1)$$\mathrm{OUR}\;=\frac{F_{\mathrm{in}}}{V}*V_{\mathrm{mol}}-\left[x_{O2,\mathrm{in}}-\left(\frac{1-x_{O2,\mathrm{in}}-x_{\mathrm{CO}2,\mathrm{in}}}{1-x_{O2,\mathrm{out}}-x_{\mathrm{CO}2,\mathrm{out}}}\right)*x_{O2,\mathrm{out}}\right]$$
(2)$$\mathrm{CPR}=\frac{F_{\mathrm{in}}}{V}*V_{\mathrm{mol}}\left[x_{\mathrm{CO}2,\mathrm{out}}*\left(\frac{1-x_{O2,\mathrm{in}}-x_{\mathrm{CO}2,\mathrm{in}}}{1-x_{O2,\mathrm{out}}-x_{\mathrm{CO}2,\mathrm{out}}}\right)-x_{\mathrm{CO}2,\mathrm{in}}\right]$$
where *F*_in_ is the inflow rate [L h^−1^], *V* is the volume of the reactor [L], *V*_mol_ is the molar gas volume (22.4 L mol^−1^), and *x_i_* is the mole fractions [%] of O_2_ or CO_2_, respectively [28].

Modeling the relationship between biomass *X* and the OUR can be achieved by a Luedeking-Piret-type equation [23]. As described by [22,29] the cumulative approach improves the signal-to-noise ratio, while masses, such as biomass and metabolic products, are better correlated:(3)$$\int_{t0}^{t}OUR\left(t\right)=\alpha *\int_{t0}^{t}X^{\prime }\left(t\right)+\beta *\int_{t0}^{t}X\left(t\right)$$
and the state estimator equation for biomass *X*_m_ is adapted from [29] when maintenance is negligible:(4)$$X_{m}≅\frac{cOUR_{m}}{\alpha }+X_{0}$$
where *X*_0_ is the initial biomass concentration, *α* is the corresponding yield coefficient [g g^−1^], and *β* is the corresponding maintenance coefficient [g g^−1^ h^−1^].

## 3. Results

The goal of this study was to characterize different *E. coli* KO strains with a model-based framework in a digitized and automated high-throughput laboratory. After performing initial screening experiments in MTPs, these strains were cultivated in our high-throughput facility in 10 mL MBRs and 100 mL small-scale bioreactors. For each strain, a parameter estimation was performed to characterize and compare the strains. Additionally, to prove the benefits of our platform-based approach, a generic state estimator for biomass (see Section 2.5) was implemented based on our model-derived parameters from the cultivations in the 2mag system to comprehensively characterize the strains. 

### 3.1. Initial Growth Characterization

To conduct preliminary investigations of growth behavior and batch-phase duration at varying glucose concentrations, the initial cultivations were performed in OxoDish MTPs with the online monitoring of DOT. These experiments were performed to obtain information for later experiments. Figure 3 displays the DOT trends during the cultivations, where each of the three strains were cultivated with different glucose concentrations: 5 g L^−1^, 10 g L^−1^, and 20 g L^−1^. For strain WG, higher glucose concentrations were associated with a longer batch phase, as seen by the sudden increase in DOT. Strain WGP demonstrated an interesting pattern, with the 20 g L^−1^ and 10 g L^−1^ glucose cultivations lasting for a similar length of time. In contrast, the 5 g L^−1^ glucose cultivation had a significantly longer lag phase but subsequently exhibited a comparable batch-phase duration to the other strains and glucose concentrations. Finally, strain WGM showed an extended lag phase at glucose concentrations of 20 g L^−1^ and 10 g L^−1^. However, this strain exhibited a similar batch-phase duration with respect to different glucose concentrations when compared to the other cultivations. All strains with 10 g L^−1^ or 20 g L^−1^ glucose ran into oxygen limitations in the Oxodish plates. For strains WG and WGP, a second decrease in DOT could still be observed at glucose concentrations of 10 g L^−1^; however, interestingly, this was not the case for the other strains or glucose concentrations. 

To study the behavior of the strains under conditions similar to an industrial scale and to obtain parameters for model-based analysis, they need to be grown in more complex cultivation systems at higher glucose concentrations. These mini-bioreactors are available in the high-throughput facility of the lab in the chair of bioprocess engineering, which utilizes the 2mag system, as described in Section 2.2.2. To evaluate the strains’ robustness towards high glucose concentrations, the different strains were cultivated under high initial glucose concentrations and pulsed once (in the case of a 10 g L^−1^ initial glucose concentration) or twice (in the case of a 5 g L^−1^ initial glucose concentration) to their initial glucose concentration. This assessment aims to determine the strains’ resilience towards varying initial glucose and biomass concentrations and to determine whether the cells experience oxygen limitations or whether knockout limits glucose uptake rates. This could limit the OURs to prevent anoxic conditions and the resulting overflow or anoxic metabolism.

As can be seen in Figure 4, the strains WG, WGP, and WGM showed a similar growth behavior with a slightly longer lag phase for the strain WGM. The cultivations with the higher initial glucose concentration also showed higher amounts of biomass at the end of the batch phase. Furthermore, higher amounts of acetate were also measured in the cultivations with the highest glucose concentration (10 g L^−1^). After the cultivations were brought back to their initial glucose level by the glucose pulse, the cells were shown to partially run into oxygen limitations and there was higher acetate formation. Overall, at the end of cultivation, a similarly high biomass could be observed for all three strains, with WG showing the highest q_Smax_ and Y_X/S,em_ (Table 2). A complete overview of the parameters is given in Table 2.

### 3.2. BioXplorer Cultivation

Strain characterization in the 2mag system (Section 2.2.2) revealed that WG and WGP exhibited the best trade-off between the high substrate uptake rates q_Smax_ of 1.13 and 0.97 g g^−1^ h^−1^, respectively, and the low acetate production rate q_Apmax_ (0.10 and 0.04 g g^−1^ h^−1^) at an initial glucose concentration of 10 g L^−1^. Hence, WG and WGP were cultivated in the parallel small-scale BioXplorer system with the capability for higher OTRs and an additional online off-gas analysis (Section 2.2.3). Therefore, an initial glucose concentration of 20 g L^−1^ was selected for further testing. Additionally, the implementation of a biomass estimator based on stoichiometric parameters from the 2mag system was assessed and the Root Mean Square Errors (RMSEs) of the estimations were compared.

As depicted in Figure 5, the O_2_ content in the off-gas environment steadily decreased while the CO_2_ content increased for both strains. No oxygen limitation occurred, as seen previously in the Oxodish MTPs (Figure 3). Correspondingly, the OURs and CPRs increased up to 140 mmol_O2_ L^−1^ h^−1^ and 143 mmol_CO2_ L^−1^ h^−1^ for WG and 109 mmol_O2_ L^−1^ h^−1^ and 118 mmol_CO2_ L^−1^ h^−1^ for WGP, respectively. The peaks in the OUR and CPR signals at approximately 2.8 h and 6.8 h were due to increases in the aeration rate. The biomass concentration steadily increased while the glucose concentration steadily decreased (Figure 5), with a q_Smax_ of 1.03 g g^−1^ h^−1^ and 0.98 g g^−1^ h^−1^ for WG and WGP, respectively. Acetate accumulation due to overflow metabolism and acetate recycling (reuse of produced acetate by the same cells as the carbon source) occurred for both strains. Interestingly, as shown in Figure 5, WG and WGP began to consume acetate, even when glucose was present in the medium. This behavior is more pronounced in strains WG and WGP and cannot be clearly observed for WGM, as demonstrated in Figure 4. WG showed a decrease in q_Smax_, q_Apmax_, q_Omax_, and Y_XSem_ while WGP exhibited a similar q_Smax_ but a reduction in Y_XSem_ and q_Omax_ compared to the cultivations in the 2mag (Table 2). The maintenance coefficient stayed similar for all cultivations, except for WG with 10 g L^−1^ initial glucose. Interestingly, the glucose affinity based on the model is also very similar for all strains.

Generic biomass of cumulative OUR (Section 2.5) was implemented. Instead of using an optimization algorithm for the offline determination of stochiometric parameters, the estimated parameters (Table 2) from the 2mag experiments were chosen. Afterwards, the estimation was updated offline with the corresponding parameters from the BioXplorer cultivation (solid line). The RMSEs for the initial biomass estimation were 0.64 g L^−1^, 0.77 g L^−1^, 0.47 g L^−1^, and 0.48 g L^−1^, after updating the stoichiometric parameters offline for WG and WGP, respectively.

## 4. Discussion

In this study, we characterized three isogenic *E. coli* KO strains with impaired glucose uptake capacities using our robotic HT screening facility. We used a combination of MTPs, MBRs, and STRs to avoid the limitations of a single cultivation platform and demonstrate the advantages of an automated platform-based approach. Our initial MTP experiments showed that strains with 20 g L^−1^ glucose went into oxygen limitation due to limitations in the OTRs of 30 mmol_O2_ L^−1^ h^−1^ under the applied conditions [30]. This highlighted the need for lower glucose concentrations to prevent anoxic conditions and bias in the analysis. Overall, our study provides valuable insights into the early phase of microbial bioprocess development and the trade-offs between throughput, limited OTRs, and analytics required for a comprehensive strain characterization [15,30].

Consequently, the strains were cultivated with 5 g L^−1^ or 10 g L^−1^ initial glucose in 10 mL MBRs with additional atline analytics for biomass, glucose, and acetate. This closed the gap of limited analytics while maintaining high throughput. The strain WG showed the shortest batch phase, which was expected as it has only one KO in the PTS. The strains WGP and WGM have longer batch phases, which can be attributed to the additional KO in the EII glucose-specific PTS component or that of GalP, respectively [31]. Their glucose uptake is further impaired, leading to the determined lower q_Smax_, which is then reflected in their lower growth rates [32], which subsequently resulted in lower OURs. Estimated values for the glucose affinity K_S_ were much larger than reported for the wild-type strain (0.003 g L^−1^) [33], which can be attributed to the inactivated PTSs in these cells [9]. However, fitting this parameter is challenging and its significance is only observed when the glucose concentration is not significantly greater than the K_s_. However, the model-based estimation of this parameter is a much simpler way to estimate this value than, for example, using a chemostat or using microfluidic systems, as has been shown recently [34]. Moreover, the results are notable as they indicate that there was no reduction in Y_X/S,em_ when the initial glucose concentration was increased from 5 g L^−1^ to 10 g L^−1^. This is attributed to the lower acetate production, which prevents energy wastage through overflow pathways [35]. This highlights that the characterized strains are of particular interest when cultivated under high glucose concentrations and producing less acetate, which can also be beneficial for the production of recombinant proteins [36]. Cultures were re-fed either once (in the case of a 10 g L^−1^ initial glucose) or twice (in the case of a 5 g L^−1^ initial glucose) to their initial glucose concentration to examine the effect of high glucose concentrations at different cell densities. In general, WGM showed the highest acetate production, with 5 g L^−1^ initial glucose. This contrasts with the results of Fuentes et al., who reported acetate production for WG and WGP and no acetate production for WGM in shake flasks with 2.5 g L^−1^ initial glucose [9]. However, Fragoso-Jiménez et al. later reported similar acetate production to the example in this study for WG and WGM; the experiment was conducted in shake flasks and in a bioreactor cultivation with 10 and 20 g L^−1^ glucose, respectively [10]. The findings of Fragoso-Jiménez can, thus, probably be attributed to the low glucose concentration used. Moreover, Steinsiek et al. reported that *ptsG* single, double, and triple KO mutants can still produce low amounts of acetate [32], supporting the results in this study. For further characterization, WG and WGP showed the best compromise between a high q_Smax_, a low q_Apmax_ and a low lag phase. Additionally, it was also possible to compare whether a single or double KO is more resistant to high glucose concentrations. Parallel small-scale systems with off-gas analyses were employed to cultivate WG and WGP strains in 100 mL STRs. This was conducted because the accurate determination of the OUR and q_Omax_ is of great interest, especially at higher glucose concentrations where the oxygen supply is critical [37]. These systems offer a reasonable trade-off between throughput and additional process analytics.

Hence, WG and WGP were cultivated in 100 mL STRs equipped with off-gas analyses. Such parallel small-scale systems offer a reasonable trade-off between throughput and additional process analytics. Oxygen-related parameters, such as q_Omax_ and Y_OX_, were derived from mass balances based on off-gas measurements. The OUR and the respective q_Omax_ showed an expected reduction compared to previously determined values for the parental strain W3110 [5,38]. For WG and WGP, Y_X/Sem_ was diminished, indicating a waste of carbon in the cells when cultivated under high glucose concentrations, as already reported by Lara et al. [8]. Since q_Apmax_ and q_m_ had not increased, this loss could be attributed to other side products as the production of lactate, ethanol, and succinate could remain unaffected [32]. It is also interesting to see that all mutants consume acetate with a similar efficiency, which would be an important aspect if these strains would be applied in high cell density fed-batch processes. The consumption of acetate by WG and WGP, while glucose was still present in the medium, is contrary to the expected diauxic growth behavior due to catabolite repression [39]. However, O’ Beirne and Hamer [40] have reported the co-consumption of glucose and acetate for the parental strain W3110 when grown at low growth rates in continuous cultures. A recent study has also demonstrated that *E. coli* is capable of co-consuming glucose and acetate under glucose excess [41]. The impaired glucose uptake in WG and WGP leads to lower uptake rates and is likely to lower intracellular glucose levels. Thus, increased cAMP levels can occur, which, in turn, relieves the repression of the genes involved in the utilization of alternative carbon sources, such as acetate, and could be an explanation for this observation. Notably, strain WGM does not show this behavior, even though it also carries a KO in the PTS. To further investigate this, glucose-acetate-switch experiments could also provide more information on a gene-expression level [42].

By using an automated cultivation platform that combines different cultivation systems, it has been demonstrated that the experimental effort for strain characterization can be reduced. The common data infrastructure of the platform also enables the implementation of mathematical tools, contributing to a quality-by-design approach at an early stage [43]. This facilitates the derivation of strain-specific parameters, leading to a more knowledge-based approach to bioprocess design. Furthermore, the timelines for the implementation of soft sensors can be shortened [44]. However, as the complexity of managing modeling and experimental workflows in such high-throughput robotic facilities increases, the implementation of sophisticated workflow management systems becomes inevitable to ensure robust and reproducible experiments [45]. These advancements pave the way for strong and reliable discussions on the future of automated bioprocess development [46].

## 5. Conclusions

In this study, we present an automated and comprehensive platform that offers a novel approach for characterizing the libraries of engineered or evolved strains with metabolic traits under well-controlled and defined conditions. This unique capability is not available in other typical formats and has enabled us to connect our platform with a microkinetic growth model, resulting in the successful estimation of relevant physiological parameters. Our findings have the potential to significantly enhance the engineering perspective of bioprocess development, bridging the gap between the upstream and downstream stages. Further developments in coupling this with product formation models and the exploration of other model organisms would broaden the applicability of our platform.

The characterization of the three *E. coli* KO strains under study could successfully be performed in several stages. The parameters found using our model-based approach were in line with findings from previous studies. The multi-KO strains showed reduced acetate production and might be used as interesting host strains in large-scale batch cultivations; they should be investigated for the production of recombinant proteins.

## Figures and Tables

**Figure 1 bioengineering-10-00808-f001:**
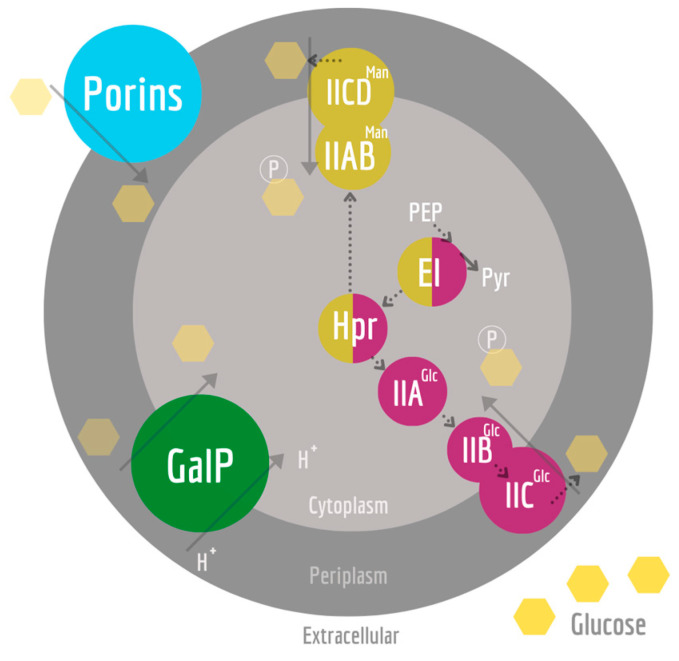
Glucose uptake systems involved in the KO strains used in this work. Glucose reaches the periplasm in a passive form through porins (blue), such as OmpC, OmpF, and LamB, or through active transport. Such active transport to the cytoplasm can occur via PEP-dependent PTS for glucose (PTS^Glc^ in purple), mannose (PTS^Man^ in gold), and proton-driven symporter GalP (in green). PEP-dependent PTS glucose transport starts when EI is phosphorylated by phosphoenolpyruvate (PEP). The phosphorylation reaches Hpr and, finally, the EII complex, where the substrate is both phosphorylated and translocated. If glucose is taken up by GalP, glucokinase phosphorylates glucose in the cytoplasm (not shown). The dotted lines indicate phosphorylation reactions.

**Figure 2 bioengineering-10-00808-f002:**
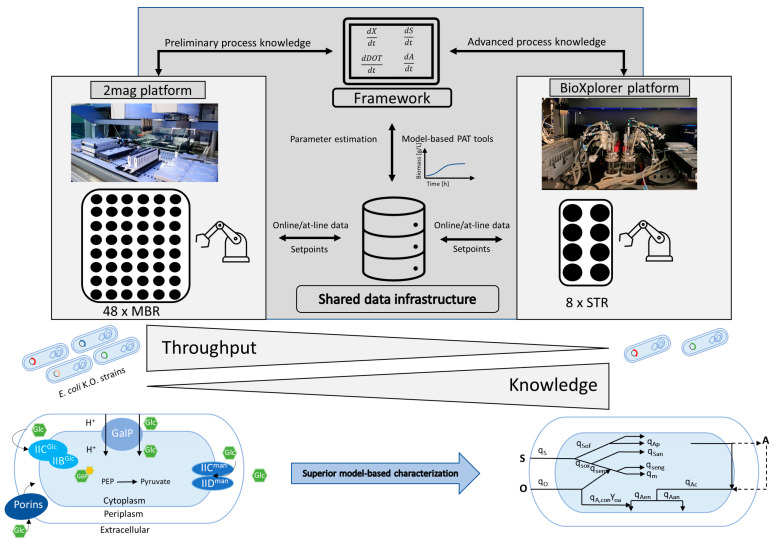
Visualization of the workflow across several cultivation platforms in a digitized and automated lab. By using different platforms, this approach offers the possibility to resolve the contradiction between throughput and knowledge gain by leveraging data from different sources and using them for further process analysis.

**Figure 3 bioengineering-10-00808-f003:**
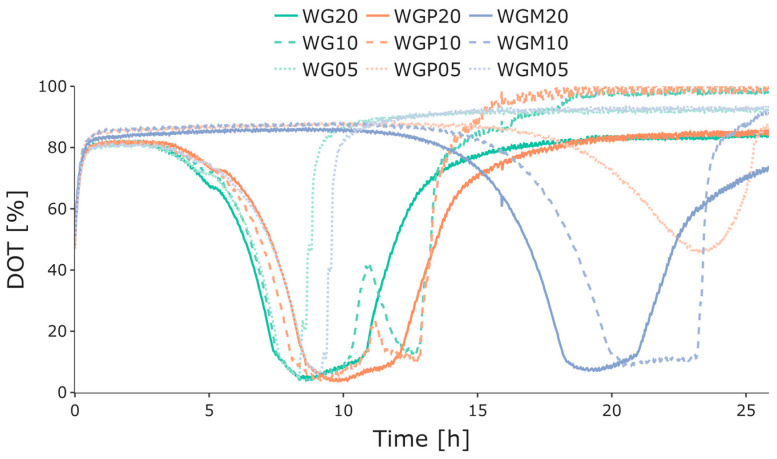
Measured online DOT values for batch cultivations of *E. coli* WG (green), WGP (orange), and WGM (blue) in 24-well OxoDish MTPs with 20 g L^−1^, 10 g L^−1^, and 5 g L^−1^ initial glucose concentrations (indicated by the number after the strain name). The differences in the length of the batch phases are highlighted by the sudden increase in DOT after glucose depletion. Strain WGM also exhibited a longer lag phase prior to growth. Dissolved oxygen tension, DOT; microtiter plate, MTP.

**Figure 4 bioengineering-10-00808-f004:**
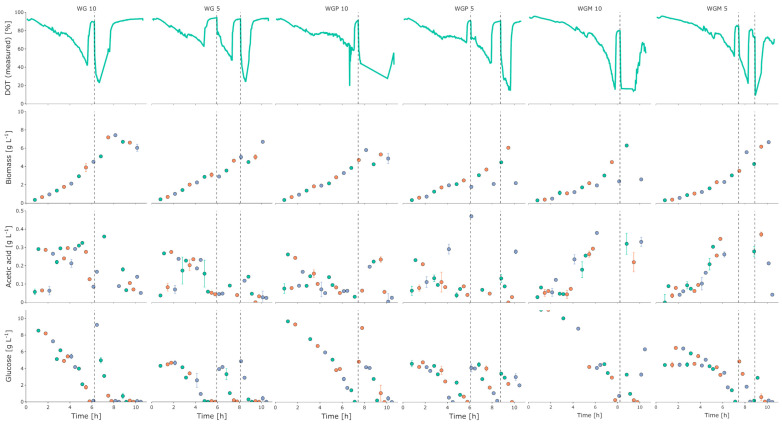
Cultivation data from strains WG, WGP, and WGM from the 2mag platform for 10 and 5 g L^−1^. Dashed vertical lines indicate glucose pulses where the cultivation was pulsed to the original glucose concentration. Different colors indicate the three replicate reactors.

**Figure 5 bioengineering-10-00808-f005:**
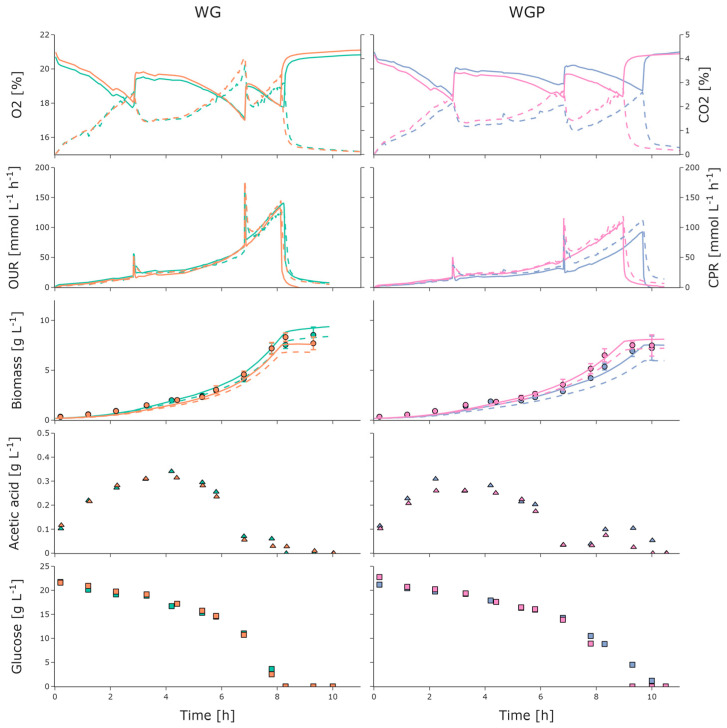
Measured online and atline values for batch cultivations of *E. coli* WG (**left**) and WGP (**right**) as biological duplicates (each color referring to one of the duplicates). Measured online values for O_2_ (solid) and CO_2_ (dashed) and corresponding online values for OUR (solid) and CPR (dashed) derived from off-gas analysis. Measured atline values for biomass (circle) and online values for biomass estimated from OUR (dashed, initial parameters from 2mag; line, updated parameters from BioXplorer), acetic acid (triangle), and glucose (square). Error bars derived from triplicates (*n* = 3). Oxygen uptake rate, OUR; carbon dioxide production rate, CPR.

**Table 1 bioengineering-10-00808-t001:** *E. coli* KOs used in this study, derived from [9]. Gene *ptsG* encodes the IIBC component of the phosphotransferase system, which mediates the uptake and concomitant phosphorylation of glucose. Gene *manX* encodes the ManX subunit of the mannose phosphotransferase system permease. Gene *galP* encodes the GalP member of the major facilitator superfamily of transporters and is a major route for galactose transport into *E. coli*.

Strain	Genotype
WG	W3110 Δ*ptsG*
WGM	W3110 Δ*ptsG*, Δ*manX*
WGP	W3110 Δ*ptsG*, Δ*galP*

**Table 2 bioengineering-10-00808-t002:** Estimated parameters and their uncertainties for the different *E. coli* strains and initial glucose conditions (number in brackets [g L^−1^]). Cultivations with 5 and 10 g L^−1^ were performed in the 2mag MBR; cultivations with 20 g L^−1^ were performed in the BioXplorer bioreactors. Maximum growth rate µ_max_, substrate affinity constant K_s_, maximum specific substrate uptake rate q_Smax_, maximum specific acetate production rate q_Apmax_, maximum specific acetate consumption rate q_Acmax_, maximum specific oxygen uptake rate q_Omax_, maintenance coefficient q_m_, yield coefficient biomass on glucose, excluding maintenance Y_X/S,em_, yield oxygen on biomass Y_OX_.

Parameter	WG (5)	WG (10)	WGP (5)	WGP (10)	WGM (5)	WGM (10)	WG (20)	WGP (20)
µ_max_ [h^−1^]	0.55 ± 0.06	0.62 ± 0.09	0.46 ± 0.04	0.45 ± 0.04	0.41 ± 0.03	0.47 ± 0.05	0.44 ± 0.03	0.36 ± 0.02
K_S_ [g L^−1^]	0.010 ± 0.002	0.009 ± 0.002	0.011 ± 0.002	0.010 ± 0.002	0.010 ± 0.002	0.011 ± 0.002	0.010 ± 0.002	0.010 ± 0.002
q_Smax_ [g g^−1^ h^−1^]	1.12 ± 0.01	1.13 ± 0.01	0.98 ± 0.01	0.97 ± 0.01	0.87 ± 0.01	0.98 ± 0.01	1.03 ± 0.01	0.98 ± 0.01
q_Apmax_ [g g^−1^ h^−1^]	0.10 ± 0.01	0.10 ± 0.01	0.05 ± 0.00	0.04 ± 0.00	0.13 ± 0.01	0.08 ± 0.01	0.06 ± 0.01	0.06 ± 0.01
q_Acmax_ [g g^−1^ h^−1^]	0.07 ± 0.00	0.07 ± 0.01	0.06 ± 0.00	0.06 ± 0.01	0.06 ± 0.01	0.07 ± 0.00	0.08 ± 0.00	0.0708 ± 0.00
q_Omax_ [g g^−1^ h^−1^]	0.79 ± 0.01	0.91 ± 0.01	0.67 ± 0.01	0.66 ± 0.01	0.62 ± 0.01	0.68 ± 0.01	0.55 ± 0.01	0.41 ± 0.01
q_m_ [g g^−1^ h^−1^]	0.047 ± 0.002	0.056 ± 0.004	0.044 ± 0.003	0.047 ± 0.004	0.044 ± 0.004	0.044 ± 0.003	0.044 ± 0.002	0.044 ± 0.003
Y_X/S,em_ [g g^−1^]	0.53 ± 0.06	0.56 ± 0.08	0.49 ± 0.04	0.49 ± 0.05	0.50 ± 0.04	0.51 ± 0.05	0.45 ± 0.03	0.39 ± 0.02
Y_OX_ [g g^−1^]	1.43 ± 0.01	1.49 ± 0.01	1.45 ± 0.01	1.47 ± 0.01	1.46 ± 0.01	1.45 ± 0.01	1.25 ± 0.01	1.2 4 ± 0.01

## Data Availability

The data presented in this study are available in the following public repo: https://git.tu-berlin.de/bvt-htbd/public/automated_strain_characterization (accessed on 9 June 2023).

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
