# Peer review of "Model-Based Characterization of E. coli Strains with Impaired Glucose Uptake"

_bioengineering, 2023, doi:10.3390/bioengineering10070808_

Round 1

Reviewer 1 Report

Report

The present research bioengineering-2470236 titled: “Characterization of E. coli strains with restricted glucose uptake in model based scale down experimentswas aimed at the characterization of three different E. coli knockout strains with limited glucose uptake capacity using automated model-based characterization: The topic is very interesting from the and molecular and biotechnology prospectives. However; there are some comments and suggestions that should be considered and fulfilled and these are as follows:

Comments

1.     The title of the manuscript differs from the online submission system “Characterization of E. coli strains with restricted glucose uptake in model based scale down experiments and the manuscript pdf file “Model-based Characterization of E. coli Strains with Impaired 2 Glucose Uptake”. So, the author should define which one is the final?. 2.     The abstract is well-written and organized with a well-written and defined conclusion. 3.     In the introduction, L47-48 “Production of acetate can inhibit further growth and represents a loss of carbon, so acetate formation should be prevented” needed relevant reference citation(s). 4.     Abbreviations should be first described at the first mention and then used consistently in the whole manuscript (examples, MBR (L73). The whole manuscript should be accordingly thoroughly revised 5.     L114, Reference source not found. 6.     Figures 1 and 2. Are they designed by authors? or of other sources/references” 7.     In the methods, L126, Reference source not found.. 8.     L127, the authors should cite these publication(s). 9.     In Table 1, the authors should describe the gene (function) at which the Knock-out mutation was established in the table Footnote. For Example, ΔptsG, ΔmanX, ΔgalP 10.  L262 & L315, L246, L353, 358, 363, Error! Reference source not found 11.  In the results there are many interpretations of the obtained results. Therefore, I recommend the transfer of all the results interpretation to the discussion section for better follow-up.. Therefore, and for the above-mentioned remarks, I advised a minor revision of the respective manuscript in its current state taking into consideration the above comments and recommendations before being considered for publication

Reviewer 2 Report

In the manuscript entitled "Model-based Characterization of E. coli Strains with Impaired Glucose Uptake" three different E. coli knockout strains with limited glucose uptake capacity were characterized with a model-based framework in a digitized and automated high-throughput laboratory. This platform offers a novel approach for characterizing libraries of engineered or evolved strains with metabolic traits under well-controlled and defined conditions. The results of this study have the potential to significantly enhance the engineering perspective of bioprocess development, bridging the gap between upstream and downstream stages. Therefore, this study has a merit for publication in the Bioengineering.

There are various parts in the text as "Error! Reference source not found". Please revise them. Also, the sentence in the line 304 is incomplete.

Reviewer 3 Report

This manuscript by Krausch et al. studied the characterization of E. coli strain, with deletion in sugar transport system, regarding to glucose metabolism. Three different cultivation platform are utilized for the characterization. The authors used the properly design experiments to meet the objective of the study. Paper possesses good amount of contents and information. With that being said, some work still needed to be polished to improve the overall quality of paper. In addition, there are some unbalanced sentences, grammatical errors and typos giving a bumpy flow to the reading. The authors are strongly encouraged to correct these errors for a cohesive and concise presentation of this interesting work. Thus, the reviewer thought this paper need minor revision before considering for the publication in Bioengineering.

Major:                                                  

Comment 1: Authors are strongly suggested to clearly state the scope and aim of this study, and approaches followed in last paragraph of introduction section.

Comment 2: Authors used 3 different cultivation platform for culturing the E. coli strains (both wild type and engineered strain) and stated that the results obtained in each cultivation platform is comparable. Authors are strongly suggested to provide a table showing results of all cultivation platform together for easy comparisons.

Comment 3: Authors showed very interesting results that E. coli WG and WGP strain co-consumed acetate and glucose after 4 hours of incubation whereas E. coli WGM did not showed such characteristic. As acetate is a very poor carbon source, why only two E. coli strain (WG and WGP) prefer acetate even in presence of higher glucose concentration. Authors are encouraged to discuss the observed phenomenon.

Minor:

Comment 1: Authors are suggested to rewrite Line 14-15. It’s difficult to understand what authors want to highlights.

Comment 2: Figures and tables are not linked to the text. Authors are suggested to link the figure to text.

Comment 3: In Line 126, authors are suggested to change the word “project” to “study”.

Comment 4: Authors are suggested to remove line 127.

Comment 5: Line 155, authors mentioned about the measurement of fluorescence. But no result regarding to fluorescence was shown. Authors are suggested to rectify the mistake.

Comment 6: Authors are suggested to remove Line 250-257. Authors are suggested to discuss directly result in result section.

Minor editing is needed
